# More than Garden Variety: Massive Vegetations from Infective Endocarditis

**DOI:** 10.3390/pathogens9120998

**Published:** 2020-11-29

**Authors:** Christopher Radcliffe, Joyce Oen-Hsiao, Matthew Grant

**Affiliations:** 1Yale School of Medicine, New Haven, CT 06510, USA; joyce.oen-hsiao@yale.edu (J.O.-H.); m.grant@yale.edu (M.G.); 2Section of Cardiovascular Medicine, Department of Internal Medicine, Yale New Haven Hospital, New Haven, CT 06510, USA; 3Section of Infectious Diseases, Department of Internal Medicine, Yale New Haven Hospital, New Haven, CT 06510, USA

**Keywords:** infective endocarditis, vegetation, tricuspid valve

## Abstract

Infective endocarditis classically involves non-sterile vegetations on valvular surfaces in the heart. Feared complications include embolization and acute heart failure. Surgical intervention achieves source control and alleviates valvular regurgitation in complicated cases. Vegetations >1 cm are often intervened upon, making massive vegetations uncommon in modern practice. We report the case of a 39-year-old female with history of intravenous drug abuse who presented with a serpiginous vegetation on the native tricuspid valve and methicillin-resistant *Staphylococcus aureus* bacteremia. The vegetation grew to 5.6 cm by hospital day two, and she successfully underwent a tricuspid valvectomy. Six weeks of intravenous vancomycin therapy were completed without adverse events. To better characterize other dramatic presentations of infective endocarditis, we performed a systematic literature review and summarized all case reports involving ≥4 cm vegetations.

## 1. Introduction

Infective endocarditis classically involves non-sterile vegetations on valvular surfaces in the heart [1]. Feared complications include embolization, uncontrolled sepsis, destruction of valvular tissue, and acute decompensated heart failure [1]. Current recommendations [2,3,4,5] endorse surgical intervention when certain criteria are satisfied and vegetations exceed 1 cm in length. 

We report the case of a 39-year-old female with history of intravenous drug abuse (IVDA) who presented with a serpiginous vegetation on the native tricuspid valve and methicillin-resistant *Staphylococcus aureus* (MRSA) bacteremia. The vegetation grew to 5.6 cm, and she underwent a tricuspid valvectomy. Six weeks of intravenous (IV) vancomycin were completed without adverse events. We performed a systematic literature review and summarized all case reports involving ≥4 cm vegetations.

## 2. Case Presentation

A 39-year-old female with a history of IVDA and untreated hepatitis C presented to the emergency department with a two-week history of malaise, arthralgias, and myalgias accompanied by a non-productive cough. In the preceding years, she had multiple admissions for MRSA soft tissue abscesses, MRSA septic arthritis, MRSA pneumonia, and one episode of MRSA bacteremia. In the emergency department, she was febrile (40 °C, rectal) and tachycardic. No peripheral stigmata of endocarditis were noted on physical exam. Blood cultures were obtained before empiric IV vancomycin and piperacillin-tazobactam were started. 

On admission, laboratory investigations were notable for hyponatremia (126 mmol/L; ref. range 136–144 mmol/L) and an absence of leukocytosis (9900/μL; ref. range 4000–10,000/μL). Transthoracic echocardiography revealed tricuspid regurgitation and a large, serpiginous vegetation on the septal leaflet of the tricuspid valve. Additionally, a chest radiograph demonstrated patchy opacities in multiple lobes, which were concerning for septic emboli. 

Blood cultures grew MRSA the following day, and antimicrobial therapy was narrowed to IV vancomycin. A transesophageal echocardiogram showed progression to severe tricuspid regurgitation with all three leaflets affected by destructive vegetations. The vegetation on the septal leaflet had grown to 5.6 cm and extended into the right atrium (Figure 1). She was transferred to our institution’s primary hospital for surgical management.

On hospital day three, she underwent tricuspid valvectomy. A median sternotomy was performed. Ascending aorta and bicaval cannulation were carried out for cardiopulmonary bypass. The right atrium was opened through an oblique atriotomy. Extensive destruction of the tricuspid valve was noted intraoperatively, and all leaflets were excised and debrided down to the tips of the papillary muscles. The annulus was also debrided. Cardiopulmonary bypass was weaned without complication. Stainless-steel wires were used to close the sternum, with absorbable sutures for skin and subcutaneous tissues. Debrided tissue was submitted for culture and subsequently grew MRSA.

Postoperatively, she required vasopressor support and remained intubated. Her leukocytosis (19,000/μL) peaked on hospital day four, so IV piperacillin-tazobactam was restarted. Blood cultures taken the same day returned positive for MRSA. By hospital day six, she was extubated and weaned from vasopressor support. Antimicrobial therapy was again narrowed to IV vancomycin. Blood cultures taken on hospital day seven returned negative, and a treatment plan to receive six-weeks of IV vancomycin starting from this date was made. Imaging studies in the following days demonstrated cavitary lesions in both lungs (Figure 2) and possible C4–C5 vertebral discitis and osteomyelitis; however, no further surgical interventions were required. She was discharged on hospital day 18.

As an outpatient, she was seen 28 days after discharge and had been tolerating vancomycin. She reported resolution of her constitutional symptoms, yet she continued to experience diffuse arthralgias and joint swelling. Her six-week course of vancomycin was completed five days later, and she went on to be diagnosed with seronegative rheumatoid arthritis in the coming months. 

The review of medical records was conducted in accordance with the Declaration of Helsinki. The protocol was approved by the Institutional Review Board of Yale University (2000028148), and need for informed consent was waived.

## 3. Discussion

Our case concerned a 39-year-old female with a history of IVDA who presented with infective endocarditis and MRSA bacteremia. A 5.6 cm serpiginous vegetation was visualized on her tricuspid valve, and she successfully underwent tricuspid valvectomy followed by vancomycin monotherapy for six weeks. Large vegetations are associated with higher morbidity and mortality [2,3,4,5], making our case’s successful outcome striking. 

To identify all case reports of infective endocarditis in adults with vegetations ≥4 cm, we searched PubMed database using the following operators: [large OR largest OR big OR biggest OR massive OR longest OR enormous OR huge OR exceptional] AND [vegetation OR vegetations] AND [valve] AND [endocarditis]. Only case reports in the English language were included. Table 1 summarizes all reported cases and our case. The mean age was 46 years, and 52% were male. The mean vegetation length was 5.5 cm (range 4–10 cm), and 39% of cases involved native tricuspid valves. Gram-positive bacteria accounted for most cases (14 of 23), and fungal pathogens were identified in 5 of 23 cases. Surgical intervention occurred in 16 of 23 cases, with percutaneous catheter-based intervention used in one case [6]. Length of targeted antimicrobial therapy ranged from <1–10 weeks. Overall mortality was 43% (9 of 21).

## 4. Conclusions

Our review of four decades of literature demonstrates that ≥4 cm vegetations are exceptional. Overall, advancements in echocardiography and surgical technique have made the diagnosis and management of infective endocarditis more standardized [1]. Nonetheless, massive vegetations are possible and benefit from a multidisciplinary approach to management.

## Figures and Tables

**Figure 1 pathogens-09-00998-f001:**
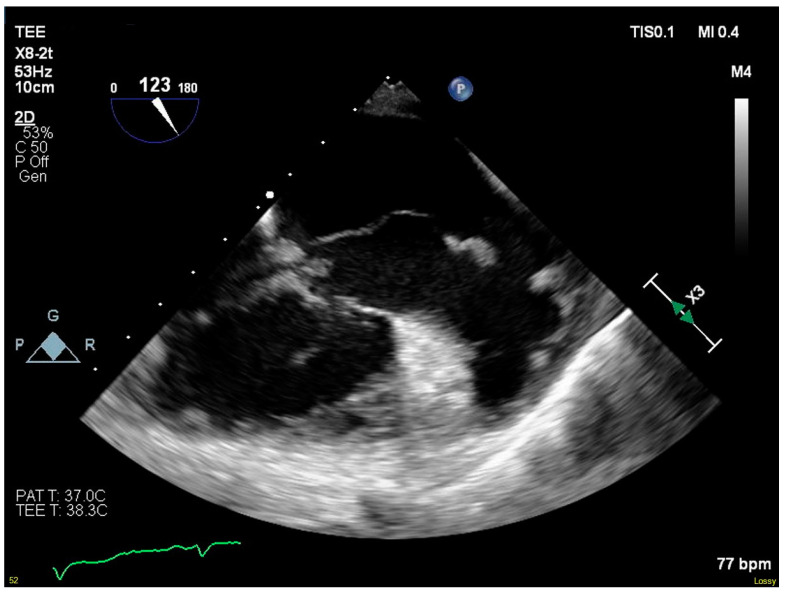
Serpiginous vegetation on septal leaflet of tricuspid valve.

**Figure 2 pathogens-09-00998-f002:**
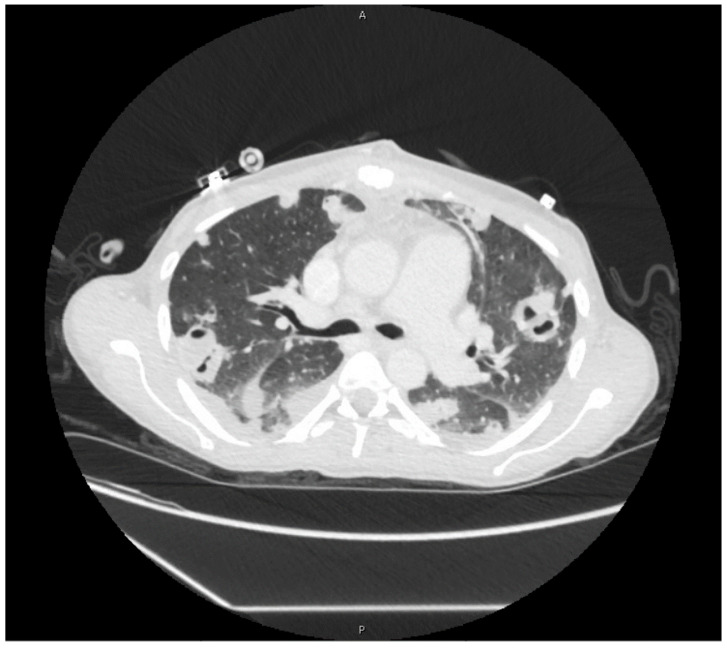
Cavitary lesions on computed-tomography scan of chest.

**Table 1 pathogens-09-00998-t001:** Summary of infective endocarditis reports with ≥4 cm vegetations.

Year and Location of Report	Age/Sex	Pathogen	Area of Involvement	Maximum Length of Vegetation (cm)	Surgical Intervention	Pathogen-Specific Antimicrobial Therapy	Length of Pathogen-Specific Antimicrobial Therapy	Outcome
2006/Canada [7]	71/M	*Aspergillus fumigatus*	pacemaker lead, superior vena cava stent	6	none	empiric only	not reported	death
1989/Japan [8]	22/F	*Candida albicans*	native tricuspid	4	vegetectomy, tricuspid valve debridement	amphotericin B → amphotericin B/miconazole/5-fluorocytosine	not reported	success
2014/USA [9]	60/F	*C. albicans*	ICD lead	4.5	ICD extraction, vegetectomy	micafungin → fluconazole	2 weeks → 6 weeks	relapse
2010/USA [10]	74/F	*Candida kefyr*	native mitral	7	none	micafungin → fluconazole	10 days → 6 weeks	success
1980/USA [11]	62/F	*Petriellidium boydii*	pacemaker lead	4	none	not reported	not reported	death
Our case	39/F	MRSA	native tricuspid	5.6	tricuspid valvectomy	vancomycin	48 days	success
2007/Japan [12]	64/M	MSSA	prosthetic mitral	7	none	cefazolin/gentamycin	<1 week	death
2010/Brazil [13]	44/M	MSSA	coronary sinus	4	vegetectomy, drainage of pyopericardium	oxacillin	not reported	success
2011/Poland [14]	20/M	MSSA	right ventricular free wall	5	vegetectomy	fluoroquinolone → vancomycin	not reported	success
1986/Spain [15]	22/M	*Staphylococcus aureus*	native tricuspid	7	vegetectomy	cloxacillin sodium/tobramycin	≥6 weeks	success
1990/Switzerland [16]	24/F	*S. aureus*	native tricuspid	4	vegetectomy, tricuspid valvuloplasty	flucloxacillin/gentamycin	56 days	success
2002/Canada [17]	30/F	*S. aureus*	native tricuspid	4.7	tricuspid valvectomy, right pulmonary artery thromboendarterectomy	not reported	not reported	success
2019/USA [18]	37/F	*S. aureus*	native mitral and tricuspid	10	none	none	none	death
1996/Japan [19]	77/M	gamma-*Streptococcus*	native tricuspid	5	tricuspid valve replacement	penicillin G/gentamycin → penicillin G/cefotiam	6 weeks → 1 month	success
2013/Taiwan [20]	81/F	*Streptococcus agalactiae*	native mitral	4.2	mitral valve replacement	penicillin G	not reported	death
2016/Japan [21]	53/M	*S. agalactiae*	native tricuspid	4	tricuspid valve replacement	not reported	not reported	not reported
2018/Portugal [22]	37/M	*Streptococcus mitis*	prosthetic pulmonic	9	pulmonic valve replacement	amoxicillin/gentamycin → vancomycin/gentamycin	4 days → 5 weeks	success
2017/USA [6]	33/F	*Streptococcus pyogenes*	native tricuspid	4.2	percutaneous extraction of vegetation	penicillin G/clindamycin	6 weeks	success
1985/Israel [23]	47/M	*Streptococcus viridans*	native mitral	4	mitral valve replacement	vancomycin	15 days	death
1977/USA [24]	57/M	*Aggregatibacter actinomycetemcomitans*	native aortic	5	aortic valve replacement	cefalotin	not reported	death
1979/USA [25]	51/M	*Haemophilus parainfluenzae*	native mitral	4.5	none	ampicillin → chloramphenicol	6 weeks → 9 days	death
1977/USA [26]	82/M	*Escherichia coli*	native mitral	4	none	ampicillin → cefalotin → cefalotin/gentamycin	2 days → 7 days → 5 days	death
2016/Italy [27]	43/F	*Proteus mirabilis*	prosthetic tricuspid	9	tricuspid valve re-replacement	not reported	not reported	not reported

Abbreviations: ICD, implantable cardioverter-defibrillator; MSSA, methicillin-sensitive *Staphylococcus aureus*; MRSA, methicillin-resistant *Staphylococcus aureus*; the arrow symbol ( → ) signifies “followed by.”.

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
