# Peer review of "More than Garden Variety: Massive Vegetations from Infective Endocarditis"

_pathogens, 2020, doi:10.3390/pathogens9120998_

Round 1
Reviewer 1 Report
1.- Degree of interest: HIG
The manuscript is of interest because the authors describe a very interesting case report, accompanied by a systematic literature review and summarized all case reports involving >4 cm vegetations and the other dramatic presentations of infective endocarditis.
2.- PRESENTATION
The description of the case report is well expressed in the text, as well as the images of the complications of infective endocarditis .The presentation, language, figure and table are adequate, clear and educational
3.- DISCUSSION
The authors make a detailed discussion with an comprehensive summary of the cases reported with vegetations> 4 cm , detailing the year and location of the report, the age of the patient, the pathogen, the area of involvement, the Maximum length of vegetation (cm), the surgical intervention, the pathogen-specific antimicrobial teraphy, and outcome
4.- BIBLIOGRAPHY
It is adequate, up-to-date
Reviewer 2 Report
The manuscript “More than Garden Variety: Massive Vegetations from Infective Endocarditis“ by Radcliffe et al. describes a case report of a 39y old patient, who is an i.v. drug abuser and who presents with a large vegetation of the tricuspid valve, which enlarged during the first 24h within the hospital and which was due to MRSA. The patient had a history of earlier MRSA abscesses, arthritis and pneumonia. She underwent surgery of her valve and after some early complications she was discharged on day 18 with further vancomycin treatment.
Although the vegetation is extremely large, there is nothing special in this case.
Reviewer 3 Report
The manuscript "More than Garden Variety: Massive Vegetations from Infective Endocarditis" reports the case of infective endocarditis,presenting huge vegetation on the tricuspid valve.The case is presented thoroughly,including bacteriological data,imaging investigations,medications and surgery.The authors also presented a systematic literature review and summarized 23 found case reports involving >4 cm vegetations,describing demographic data,pathogen,area of involvment, length of vegetation,sugery type, antibiotic treatment and outcome. Presented data is interesting and widens the understanding of complicated and uncommon presentation of infective endocarditis.
